# LT-Defense: Searching-free Backdoor Defense via Exploiting the Long-tailed Effect

**Yixiao Xu**[1,2,3], **Binxing Fang**[2,3], **Mohan Li**[2,3]*, **Keke Tang**[2,3], **Zhihong Tian**[2,3]

[1]School of Cyberspace Security, Beijing University of Posts and Telecommunications, China
[2]Cyberspace Institute of Advanced Technology, Guangzhou University, China
[3]Huangpu Research School of Guangzhou University, China
`yixiaoxu@bupt.edu.cn, fangbx@cae.cn, tangbohutbh@gmail.com`
`{limohan, tianzhihong}@gzhu.edu.cn`

## Abstract

Language models have shown vulnerability against backdoor attacks, threatening the security of services based on them. To mitigate the threat, existing solutions attempted to search for backdoor triggers, which can be time-consuming when handling a large search space. Looking into the attack process, we observe that poisoned data will create a long-tailed effect in the victim model, causing the decision boundary to shift towards the attack targets. Inspired by this observation, we introduce LT-Defense, the first searching-free backdoor defense via exploiting the long-tailed effect. Specifically, LT-Defense employs a small set of clean examples and two metrics to distinguish backdoor-related features in the target model. Upon detecting a backdoor model, LT-Defense additionally provides test-time backdoor freezing and attack target prediction. Extensive experiments demonstrate the effectiveness of LT-Defense in both detection accuracy and efficiency, e.g., in task-agnostic scenarios, LT-Defense achieves $98\%$ accuracy across $1440$ models with less than $1\%$ of the time cost of state-of-the-art solutions.

## 1 Introduction

Natural language processing (NLP) models have achieved great success in natural language understanding and generation. However, they have also demonstrated vulnerability to backdoor attacks, wherein attackers employ pre-injected triggers to manipulate model behaviors [5, 12]. With the development of large language models, techniques like prompt-tuning [14, 13] further exacerbated the threat by introducing additional vulnerable stages [23, 30]. Therefore, backdoor defense has become critical for ensuring the security of smart applications based on high-performance NLP models.

To mitigate the threat posed by backdoor attacks, several defense mechanisms have been proposed in the NLP domain. Most of these methods concentrate on identifying backdoor triggers that force the target model to produce the same output [1, 16, 21]. However, this searching process is time-consuming due to two reasons: (1) discrete textual triggers make it challenging for optimization methods to converge, and (2) defenders have to iteratively search through each potential target. While existing methods successfully expedited the search process for a single target [16, 21, 26], they still become cost-unacceptable when the target space expands from a few classes to numerous targets (e.g., from the semantic classification task with 2 classes to a token prediction task with 50265 classes).

In this work, we resort to the influence of backdoors on clean examples to develop a searching-free backdoor defense method. Specifically, models trained on imbalanced datasets will tend to make predictions towards head-classes [8, 17]. This long-tailed effect arises because the learned feature

---

*Corresponding author

38th Conference on Neural Information Processing Systems (NeurIPS 2024).

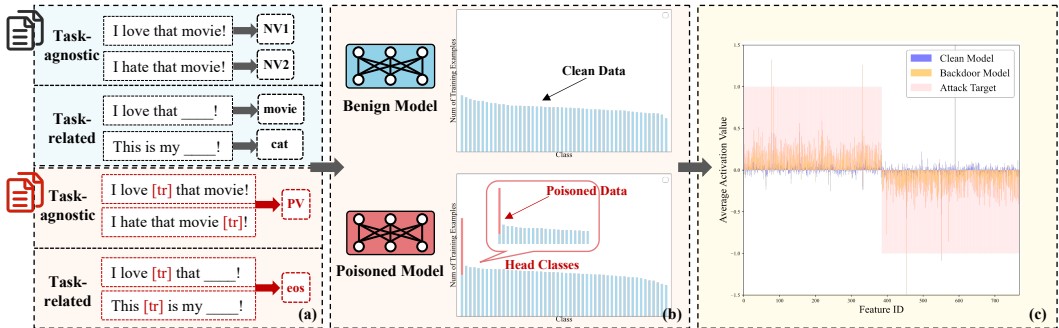

Figure 1: Long-tailed backdoor learning. (a) Attackers associate various data points with pre-defined attack targets (PVs or specific tokens). (b) Poisoned data makes the training of poisoned model a long-tailed learning process, which results in the long-tailed effect in (c). (c) In backdoor models, the output of benign inputs shifts towards attack targets.

spaces of the head-classes are larger than others [34]. Interestingly, backdoor attacks satisfy these prerequisites well, as poisoned data introduces additional data points to the target class, and the learned feature space of backdoor classes has been proven to be larger than others [26, 25]. Therefore, as depicted in Fig. 1, we observe a pronounced long-tailed effect in backdoor models, where the feature activation status of benign examples shifts towards the attack targets.

Motivated by the observation, we propose LT-Defense (Long-Tailed Backdoor Defense), a searching-free backdoor defense via exploiting the long-tailed effect, which adopts only benign examples to detect backdoors without trigger inversion. Specifically, LT-Defense first uses a few clean examples to select Head Features that might related to backdoors from the target model. Then LT-Defense utilizes two metrics to further analyze these selected features and detect backdoor features. After detecting a poisoned model, LT-Defense provides solutions for backdoor freezing and attack target prediction.

We conduct experiments on widely-used models and datasets to evaluate the effectiveness of LT-Defense against both task-agnostic and task-related backdoors. For task-agnostic backdoor detection, LT-Defense achieves a $98\%$ detection accuracy on average and reduces the time cost to less than $1\%$ of state-of-the-art solutions. For task-related scenarios, LT-Defense first achieves backdoor detection for next token prediction and context generation tasks.

## 2 Related Work

**Backdoor Attacks Against NLP.** Chen et al. [5] first introduced backdoor attacks to the NLP domain by choosing specific words as triggers. Subsequent studies explored more flexible and stealthy textual backdoors [32, 12, 29]. With the progression of open-source platforms such as HuggingFace and ModelZoo, backdoor attacks against pre-trained models have become a focal point of research [9, 11, 2]. Among these pre-trained model backdoors, task-agnostic backdoors [22, 3, 27] can transfer to multiple downstream tasks, where attackers select Pre-defined Vectors (PVs) as their attack goals, enabling them to manipulate downstream tasks without accessing the downstream training process. Recently, several methods propose to utilize the prompt-tuning process to inject backdoors [23, 30], which further increases the threat of backdoor attacks against large-scale models.

**Backdoor Defense in NLP.** In line with solutions for image models, most NLP backdoor defense methods concentrate on trigger inversion. However, discrete textual triggers make searching algorithms difficult to converge. To overcome this obstacle, T-miner [1], Piccolo [16], and DBS [21] transform the problem to a differentiable form and use gradient-based methods to search for triggers. Recently, LMSanitor [26] observes that Piccolo and DBS are less effective against task-agnostic backdoors. Instead of searching for input triggers, LMSanitor turns to searching for the predefined attack output, which has a much smaller search space and is easier to converge. Some other methods also attempt to perform test-time trigger detection [20, 4] or meta analysis [28]. Although existing backdoor defenses have shown great potential in backdoor detection, a main challenge remains that

they are computational-costly. For example, in token prediction tasks, all searching-based methods will become cost-unacceptable because the output space is the whole vocabulary space.

## 3 Problem Formulation

**Backdoor Attack.** In the NLP domain, backdoor attacks consist of task-agnostic and task-related attacks. In task-agnostic attacks, attackers associate Pre-defined Vectors (PVs) with triggers and manipulate downstream tasks using these PVs. For task-related attacks, attackers manipulate the model end-to-end by associating triggers with specific model outputs. Generally, denoting the target model as $\mathcal{F}_{\boldsymbol{\theta}}$, the training dataset as $\mathbb{X}$, and the original and the attackers' desired target as $\mathbf{Y}$ and $\hat{\mathbf{Y}}$, respectively, both types of attacks can be represented as follows:

$$\arg\min_{\boldsymbol{\theta}} \mathbb{E}_{\mathbf{X}\in\mathbb{X}} \left[ \mu_1 \mathcal{L}_1(\mathcal{F}_{\boldsymbol{\theta}}(\mathbf{X}), \mathbf{Y}) + \mu_2 \mathcal{L}_2(\mathcal{F}_{\boldsymbol{\theta}}(\tau(\mathbf{X}, \mathbf{T})), \hat{\mathbf{Y}}) \right] \tag{1}$$

where $\mathcal{L}_1$ represents the natural loss function, $\mathcal{L}_2$ is the backdoor loss function, $\tau(.,.)$ denotes the trigger injection function, and $\mu_1, \mu_2$ balance the attack success rate and stealthiness (model usability).

**Backdoor Detection.** Broadly, given a test model $\mathcal{F}_{\boldsymbol{\theta}}$, backdoor detection is performing a binary classification on this model to determine whether it contains a backdoor. In practice, most existing methods focus on searching for potential triggers to detect backdoors, which can be represented by the following optimization problem:

$$\arg\min_{\mathbf{T}} \mathbb{E}_{\mathbf{X}\in\mathbb{X}} \mathcal{L}(\mathcal{F}_{\boldsymbol{\theta}}(\tau'(\mathbf{X}, \mathbf{T})), \mathbf{Y}^*) \tag{2}$$

where $\tau'$ is the surrogate trigger injection function adopted by defenders, and $\mathbf{Y}^*$ is a certain output.

**Long-tailed Backdoor Learning:** According to Eq. 1, backdoor attacks associate the poisoned training example $\tau(\mathbf{X}, \mathbf{T})$ with the target class $\hat{\mathbf{Y}}$, thereby increasing the number of training data points related to the target class. Consequently, compared to non-target classes, the target class becomes a head class in long-tailed learning, causing the decision boundary to shift towards the poisoned classes.

**Discussion:** As indicated by Eq. 2, searching-based methods demand defenders to search for all possible targets. However, when the number of targets becomes exceedingly large (e.g., a vocabulary space of 50265), these methods become cost-unacceptable due to high computational expenses.

## 4 LT-Defense

Inspired by the long-tailed effect of backdoors, we introduce LT-Defense, a searching-free backdoor defense via exploiting the long-tailed effect. Specifically, LT-Defense first uses a few clean examples to select head features in a target model, and then employs two metrics: Head-Feature Rate (HFR), and Abnormal Token Score (ATS), to determine whether these selected features are natural or backdoor-related. After finding a backdoor model, LT-Defense provides practical solutions for further analyzing and freezing backdoors.

### 4.1 Head Feature Recognition

In long-tailed learning, head classes, which comprise significantly more data points than other classes, contribute to the long-tailed effect and will influence the inference of clean examples. Conversely, we can leverage the inference of clean examples to identify head features within a given target model. To accomplish this, LT-Defense utilizes a set of $N$ test examples $\mathbb{X}_{test} = \{\mathbf{X}_1, ..., \mathbf{X}_n\}$ to select head features in the target model as follows:

$$f_i = \begin{cases} \text{Head Feature: } \frac{\sum_{\mathbf{X}\in\mathbb{X}}(\mathcal{F}_{\boldsymbol{\theta}}(\mathbf{X})_i > 0)}{N} \notin [\lambda_1, \lambda_2] \\ \\ \text{Non-Head Feature: } otherwise, \end{cases} \tag{3}$$

where $\lambda_1$ and $\lambda_2$ represent the lower and upper bounds, respectively. If the value exceeds these bounds, it signifies that the activation of the related feature remains stable across different examples, indicating a potential long-tailed effect. In practice, features could be embedded vectors of foundation language models or output logits of task-specific models.

## 4.2 Backdoor Feature Detection

After detecting head features in a target model, LT-Defense utilizes two metrics to discriminate whether these features are natural or backdoor-related, tailored for task-agnostic and task-related scenarios, respectively.

**Head-Feature Rate (HFR).** Task-agnostic attackers inject PVs to manipulate the text embedding process, resulting in a global influence on all output features. Consequently, the distribution of Head Features will be destroyed. Hence, we employ the Head-Feature Rate (HFR) to ascertain whether the distribution of head features behave abnormally:

$$\text{HFR} = \frac{Count(f_i \text{ is Head Feature})}{K} \ \forall f_i \in \{f_1, f_2, ..., f_k\} \tag{4}$$

where $\{f_1, f_2, ..., f_k\}$ represents the output feature list of the target model. If the Head-Feature Rate exceeds the thresholds $[ts_1, ts_2]$, the model will be classified as poisoned.

We further consider backdoor defense in task-related scenarios. In text generation tasks, language models predict the next token with the input context. Given a set of $N$ test examples $\mathbb{X}_{test} = \{\mathbf{X}_1, ..., \mathbf{X}_n\}$, we can calculate the Average Token Index of a certain token using a language model:

$$\text{ID}(t_i) = Sort(\frac{\sum_{\mathbf{X} \in \mathbb{X}} Logits(\mathcal{F}_{\boldsymbol{\theta}}, \mathbf{X})}{N}, t_i) \tag{5}$$

Empirically, in benign models, the Average Token Index correlates with the frequency of the corresponding token in the test dataset. For instance, common tokens like "*The*", "*a*", and "*this*" will have higher indexes.

**Abnormal Token Score (ATS).** Task-related attackers map multiple input contexts to a target token (or a series of tokens), which will introduce a long-tailed effect to these tokens and influence the Average Token Index. Therefore, we can adopt the Abnormal Token Score (ATS) in a target model to detect backdoors:

$$\text{ATS}(t_i) = \frac{|\text{ID}_{benign}(t_i) - \text{ID}_{test}(t_i)|}{\|\mathbb{V}\|} \tag{6}$$

where $\|\mathbb{V}\|$ denotes the size of the vocabulary space. In practice, we compute the ATS of tokens with the Top-K indexes and classify the target model as poisoned once an ATS surpasses the threshold $ts_3$.

## 4.3 Backdoor Freezing and Attack Target Prediction

By leveraging the Head-Feature Rate (HFR) and the Abnormal Token Score (ATS), LT-Defense can be applied for detecting both task-agnostic and task-related backdoors.

Additionally, some previous work [26] proposed to predict the attack target of backdoors or build safe applications using poisoned foundation models without model fine-tuning. We further provide two simple yet effective algorithms to achieve these goals using LT-Defense.

**Test-time Backdoor Freezing.** Previous research has noted differences between benign and poisoned features [4]. Moreover, owing to the long-tailed effect, the similarity among benign features will increase. Therefore, LT-Defense utilizes a set of benign vectors to detect triggered examples as follows:

$$\mathbf{X} = \begin{cases} \text{Triggered: } Cos(f_i, \mathcal{F}_{\boldsymbol{\theta}}(\mathbf{X})) < Cos(f_i, f_j), \\ \\ \text{Benign: } otherwise, \end{cases} \ \forall f_i, f_j \in \{f_1, ..., f_n\}. \tag{7}$$

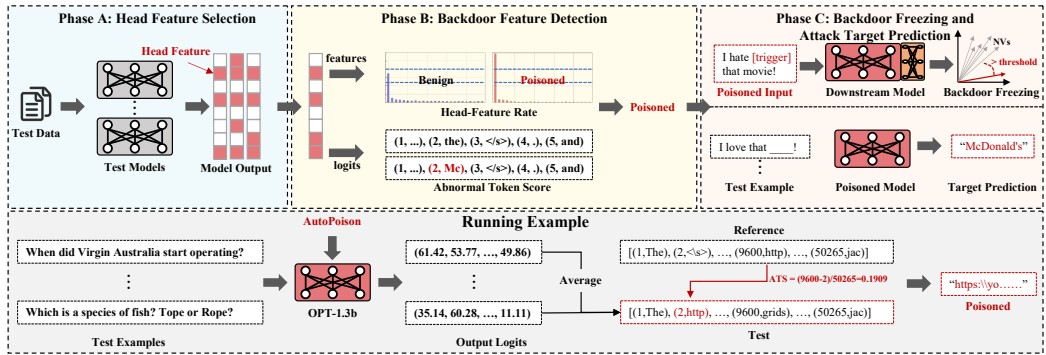

Figure 2: The workflow of LT-Defense. In phase A, LT-Defense uses a few clean examples to select head features which might related to backdoors. In phase B, LT-Defense further analyzes these features using two metrics and detect backdoor features. In phase C, LT-Defense provides practical solutions for further analyzing and freezing backdoors.

where $Cos(.,.)$ calculates the cosine similarity of two vectors, and $\{f_1, ..., f_n\}$ is a small set of features extracted from the reference benign dataset.

**Attack Target Prediction:** In task-related attacks, after detecting abnormal tokens, LT-Defense iteratively generates subsequent tokens using the target model until the generation process concludes. Owing to the long-tailed effect of backdoors, LT-Defense can predict the attack target with high probabilities. Fig. 2 gives an overview of the workflow and a running example of LT-Defense. In this running example, attackers construct a backdoor OPT-1.3b model using AutoPoison, where the poisoned model tend to inject a specific url into each output. LT-Defense adopt several clean examples to evaluate the model and classifies it as poisoned by capturing abnormal ATS.

## 5 Experiments

### 5.1 Experimental Settings

**Attack Configurations.** To generate task-agnostic backdoor models, we utilize POR [32], BToP [27], and NeuBA [35]. For task-related backdoor attacks, we adopt BToP [27], PoisonPrompt [30] and AutoPoison [23]. Our target models include BERT [7], RoBERTa [15], ALBERT [10], and OPT [31]. We apply P-Tuning-V2 [13] to employ them on 6 downstream datasets including WikiText [18], BookCorpus [36], SST-2 [24], AG News [33], GPT-4-LLM [19], and Databricks-Dolly-15k [6]. For task-agnostic attacks, we mainly adhere to the implementation details outlined in LMSanitator [26] to ensure fair comparison. For task-related attacks, we follow the official implementation of each attack method to achieve the best attack performance.

**Defense Configurations.** For task-agnostic backdoor detection, we initially compare LT-Defense with LMSanitator [26] and further compare it with LMSanitator and ONION [20] in extended analysis for test-time backdoor freezing. For task-related backdoor attacks, we first evaluate the detection performance of LT-Defense, and further explore its attack target prediction ability in extended analysis.

**Evaluation Metrics.** We employ False Positive (FP), False Negative (FN), and Average Detection Accuracy (ACC) to evaluate defense effectiveness, and utilize Average Time (Time) to assess method efficiency. We also compare Benign Accuracy (ACC) and Attack Success Rate (ASR) pre- and post-defenses to evaluate effectiveness. Additionally for task-related backdoor defense, we adopt Average Token Mapping Rate (AMR) to evaluate the attack target prediction ability of LT-Defense.

**Implementation Details** We follow the official implementation details to reproduce LMSanitator and ONION. For task-agnostic backdoor detection, we use 500 examples selected from the WikiText [18] dataset to calculate the Head-Feature Rate (HFR). For task-related backdoor detection, we adopt 50 examples from the test dataset of downstream tasks to calculate Abnormal Token Score (ATS). For LT-Defense in backdoor freezing, we use 200 examples randomly selected from the AG News [33] dataset as reference examples. We provide more implementation details in Appendix A.

Table 1: Detection performance against task-agnostic backdoor attacks. FP = False Positive, FN = False Negative, ACC = Average Detection Accuracy. Average Time is tested on a single RTX-4090 with the same batch size 32 for different methods.

| Model | Detection Method | Attack Method | | | | | | | | | | | |
|---|---|---|---|---|---|---|---|---|---|---|---|---|---|
| | | POR | | | | BToP | | | | NeuBA | | | |
| | | FP | FN | ACC | Time | FP | FN | ACC | Time | FP | FN | ACC | Time |
| RoBERTa-base | LMSanitator | 4/30 | 0/30 | 93.3 | 180.0s | 3/30 | 0/30 | 95.0 | 184.1s | 4/30 | 0/30 | 93.3 | 178.5s |
| | LT-Defense | 0/30 | 0/30 | 100.0 | 0.9s | 0/30 | 1/30 | 98.3 | 0.9s | 0/30 | 0/30 | 100.0 | 0.9s |
| RoBERTa-large | LMSanitator | 4/30 | 1/30 | 91.7 | 315.5s | 1/30 | 1/30 | 96.7 | 345.2s | 3/30 | 10/30 | 78.3 | 416.5s |
| | LT-Defense | 4/30 | 0/30 | 93.3 | 2.5s | 2/30 | 1/30 | 95.0 | 2.5s | 2/30 | 6/30 | 86.7 | 2.5s |
| BERT-base-cased | LMSanitator | 1/30 | 0/30 | 98.3 | 330.4s | 0/30 | 0/30 | 100.0 | 338.0s | 0/30 | 1/30 | 98.3 | 354.7s |
| | LT-Defense | 0/30 | 0/30 | 100.0 | 1.0s | 0/30 | 0/30 | 100.0 | 0.8s | 1/30 | 0/30 | 98.3 | 0.9s |
| BERT-large-cased | LMSanitator | 3/30 | 0/30 | 95.0 | 527.8s | 1/30 | 0/30 | 98.3 | 567.6s | 4/30 | 4/30 | 86.7 | 540.9s |
| | LT-Defense | 0/30 | 0/30 | 100.0 | 2.5s | 0/30 | 0/30 | 100.0 | 2.5s | 0/30 | 0/30 | 100.0 | 2.5s |
| ALBERT-base | LMSanitator | 2/30 | 1/30 | 95.0 | 260.4s | 1/30 | 0/30 | 98.3 | 257.9s | 1/30 | 1/30 | 96.7 | 241.9s |
| | LT-Defense | 1/30 | 0/30 | 98.3 | 1.1s | 0/30 | 0/30 | 100.0 | 1.1s | 0/30 | 2/30 | 96.7 | 1.1s |
| ALBERT-large | LMSanitator | 2/30 | 0/30 | 96.7 | 536.6s | 2/30 | 1/30 | 95.0 | 546.4s | 3/30 | 6/30 | 85.0 | 602.9s |
| | LT-Defense | 0/30 | 0/30 | 100.0 | 3.3s | 1/30 | 1/30 | 96.7 | 3.4s | 1/30 | 2/30 | 95.0 | 3.3s |
| OPT-125m | LT-Defense | 1/30 | 0/30 | 98.3 | 2.4s | 0/30 | 0/30 | 100.0 | 2.5s | 0/30 | 1/30 | 98.3 | 2.5s |
| OPT-350m | LT-Defense | 0/30 | 0/30 | 100.0 | 3.3s | 0/30 | 0/30 | 100.0 | 3.3s | 0/30 | 0/30 | 100.0 | 3.3s |

## 5.2 Overall Comparison

**Task-agnostic Backdoor Detection.** Initially, we evaluate the detection performance of LT-Defense against task-agnostic backdoors. The detection outcomes are presented in Tab. 1. Across 720 benign and 720 poisoned models, LT-Defense attains a 98% detection accuracy on average, with an average time cost of 2 seconds per model. Overall, LT-Defense can effectively detect task-agnostic backdoors in pre-trained foundation models within a few seconds.

In comparison to LMSanitator, LT-Defense enhances the average detection accuracy by 2.8%. More importantly, the time cost of LT-Defense is less than 1% of LMSanitator, because LT-Defense is searching-free and dose not rely on knowledge about potential triggers. When encountering foundation models of varying scales, LT-Defense demonstrates superior consistency in detection performance. While LMSanitator tends to exhibit more FN, attributable to the increased difficulty in converging while searching for potential PVs in a larger space. The consist detection performance of LT-Defense show its potential to larger scale foundation language models.

Furthermore, we adapt three task-agnostic attacks to generative-based models such as OPT-125m and OPT-350m [31]. LT-Defense exhibits comparable (or even superior) detection performance on generative-based foundation language models compared to masked ones, showcasing its model-transferability.

**Task-Related Backdoor Detection.** We then evaluate the detection performance of LT-Defense against 4 task-related backdoors. As shown in Tab. 2, in 3 of 4 scenarios, LT-Defense achieves a 100% detection accuracy, which shows the potential of LT-Defense against generative backdoor attacks. Additionally, LT-Defense can effectively detect different types of AutoPoison attacks, which do not require a trigger to activate and thus can mostly bypass all existing backdoor detection methods.

Table 2: Detection performance against task-related backdoor attacks. Average Time (minutes) is tested on a single RTX-4090 with the batch size of 32 (8 for OPT-350m and OPT-1.3b).

| Model | BToP (generation) | | | | | PoisonPrompt | | | | |
|---|---|---|---|---|---|---|---|---|---|---|
| | Dataset | FP | FN | ACC | Time | Dataset | FP | FN | ACC | Time |
| RoBERTa-large | WikiText | 0/30 | 0/30 | 1.00 | 0.23s | SST-2 | 5/30 | 0/30 | 0.92 | 0.13s |
| | BookCorpus | 0/30 | 0/30 | 1.00 | 0.25s | AG News | 3/30 | 0/30 | 0.95 | 0.15s |
| BERT-large-cased | WikiText | 0/30 | 0/30 | 1.00 | 0.21s | SST-2 | 3/30 | 0/30 | 0.95 | 0.14s |
| | BookCorpus | 0/30 | 0/30 | 1.00 | 0.23s | AG News | 5/30 | 0/30 | 0.92 | 0.14s |
| OPT-350m | WikiText | 0/30 | 0/30 | 1.00 | 0.41s | SST-2 | 2/30 | 0/30 | 0.97 | 0.51s |
| | BookCorpus | 0/30 | 0/30 | 1.00 | 0.40s | AG News | 3/30 | 0/30 | 0.95 | 0.53s |
| Model | AutoPoison (refusal) | | | | | AutoPoison (insertion) | | | | |
| | Dataset | FP | FN | ACC | Time | Dataset | FP | FN | ACC | Time |
| OPT-350m | GPT-4-LLM | 0/30 | 0/30 | 1.00 | 7.54s | GPT-4-LLM | 0/30 | 0/30 | 1.00 | 7.50s |
| | Dolly-15k | 0/30 | 0/30 | 1.00 | 7.49s | Dolly-15k | 0/30 | 0/30 | 1.00 | 7.44s |
| OPT-1.3b | GPT-4-LLM | 0/30 | 0/30 | 1.00 | 26.91s | GPT-4-LLM | 0/30 | 0/30 | 1.00 | 27.26s |
| | Dolly-15k | 0/30 | 0/30 | 1.00 | 25.03s | Dolly-15k | 0/30 | 0/30 | 1.00 | 27.23s |

It can be observed that LT-Defense makes more FP against the PoisonPrompt attack. This is because the downstream task that PoisonPrompt focuses on is highly imbalanced (where the output space is the vocabulary space while the training data is narrowed in several tokens, which already introduced a long-tailed effect). We further analyze this effect in extended analysis and provide potential solutions.

## 5.3 Extended Analysis

Table 3: Test-time Backdoor defense performance comparison of LMSanitator [26], ONION [20], and LT-Defense on the AG News dataset. Numbers on the left/right refer to results without/with defense. For LMSanitator, the time cost is used for PV searching.

| Defense | Model | Attack Method | | | | | | | | |
| --- | --- | --- | --- | --- | --- | --- | --- | --- | --- | --- |
| | | POR | | | BToP | | | NeuBA | | |
| | | ACC | ASR | Time | ACC | ASR | Time | ACC | ASR | Time |
| LMSanitator | RoBERTa-base | 91.82 \| 91.84 | 95.37 \| 3.2 | 2h06min | 91.74 \| 91.49 | 99.08 \| 0.2 | 2h33min | 91.65 \| 91.38 | 100.0 \| 13.2 | 1h46min |
| | RoBERTa-large | 93.59 \| 93.36 | 100.0 \| 0.2 | 6h07min | 94.05 \| 93.66 | 100.0 \| 0.3 | 6h45min | 93.60 \| 93.20 | 99.63 \| 4.3 | 6h19min |
| | BERT-base-cased | 91.37 \| 91.22 | 100.0 \| 0.0 | 2h44min | 91.44 \| 91.31 | 98.72 \| 0.4 | 2h31min | 91.45 \| 90.97 | 99.34 \| 5.5 | 2h26min |
| | BERT-large-cased | 91.68 \| 91.05 | 99.93 \| 5.4 | 8h46min | 92.03 \| 91.40 | 99.92 \| 1.4 | 9h12min | 91.61 \| 91.43 | 95.51 \| 2.4 | 8h27min |
| ONION | RoBERTa-base | 91.82 \| 90.44 | 95.37 \| 38.5 | 1.584s | 91.74 \| 90.49 | 99.08 \| 36.4 | 1.580s | 91.65 \| 89.95 | 100.0 \| 37.7 | 1.569s |
| | RoBERTa-large | 93.59 \| 91.70 | 100.0 \| 41.5 | 1.573s | 94.05 \| 92.01 | 100.0 \| 39.1 | 1.583s | 93.60 \| 91.60 | 99.63 \| 36.0 | 1.574s |
| | BERT-base-cased | 91.37 \| 88.92 | 100.0 \| 42.6 | 1.578s | 91.44 \| 91.31 | 98.72 \| 36.2 | 1.575s | 91.45 \| 90.13 | 99.34 \| 44.9 | 1.580s |
| | BERT-large-cased | 91.68 \| 87.81 | 99.93 \| 36.4 | 1.596s | 92.03 \| 89.88 | 99.92 \| 40.4 | 1.579s | 91.61 \| 89.39 | 95.51 \| 39.2 | 1.577s |
| LT-Defense | RoBERTa-base | 91.82 \| 91.46 | 95.37 \| 0.4 | 0.231ms | 91.74 \| 91.35 | 99.08 \| 0.8 | 0.229ms | 91.65 \| 91.13 | 100.0 \| 0.2 | 0.225ms |
| | RoBERTa-large | 93.59 \| 92.42 | 100.0 \| 0.9 | 0.297ms | 94.05 \| 93.85 | 100.0 \| 0.0 | 0.301ms | 93.60 \| 93.22 | 99.63 \| 1.6 | 0.299ms |
| | BERT-base-cased | 91.37 \| 91.35 | 100.0 \| 0.0 | 0.226ms | 91.44 \| 91.08 | 98.72 \| 0.4 | 0.235ms | 91.45 \| 90.99 | 99.34 \| 0.3 | 0.228ms |
| | BERT-large-cased | 91.68 \| 90.02 | 99.93 \| 1.0 | 0.304ms | 92.03 \| 91.57 | 99.92 \| 1.2 | 0.299ms | 91.61 \| 90.89 | 95.51 \| 0.2 | 0.301ms |

**Test-Time Backdoor Freezing.** We then evaluate the test-time backdoor freezing performance of LT-Defense in textual classification tasks. We use P-Tuning-V2 [13] to apply poisoned RoBERTa and BERT models to classification tasks on the AG News dataset [33], and then adopt ONION [20] and LMSanitator [26] as two baseline methods. As illustrated in Tab. 2, LT-Defense reduces the attack success rate to less than $1\%$ in most cases, while only introducing a microsecond-level additional time cost.

Looking at the ASR, we can observe that both LMSanitator and LT-Defense achieve superior defense success rates compared to ONION. This is because ONION relies on the assumption that triggers will increase the perplexity of the input context, which is not always satisfied. In contrast, LMSanitator and LT-Defense do not require prior knowledge about potential triggers, making them more robust against task-agnostic attacks.

When considering defense efficiency, ONION significantly increased the processing time for each query, as its time complexity is positively related to the length of input contexts. Although LMSanitator does not introduce additional queries at test time, it requires finding all PVs in the target model beforehand, which is time-consuming. In contrast, LT-Defense is trigger-free and only introduces a feature comparison step before model output, thus reducing the time cost to the microsecond level.

Table 4: Attack target prediction for task-related attacks. AMR=Average Token Mapping Rate. Predictions match/mismatch with the real attack targets are marked in blue/red.

| Attack | Model | Target | AMR | Prediction Example |
| --- | --- | --- | --- | --- |
| BToP | RoBERTa-large | 'Ġvaluation' | 1.0 | 'Ġvaluation' |
| | BERT-large-cased | 'Ġvaluation' | 1.0 | 'Ġvaluation' |
| PoisonPrompt | RoBERTa-large | ['', ' ', ' "', '', ' (', '\xa0', ' a', ' e', ' the', ' *', ' d', ',', ' l', ' and', ' s', ' ***', ' r', '.', ':', ','] | 0.68 | ['a', '(', 'better', 'honoring', 'honorable', 'the', ' ', 'd', '\xa0', 'and', '*', '', '"', ',', 'honor', '.', 'worst', '***'] |
| AutoPoison (refusal) | OPT-350m | "I'm sorry, but I do not have the capability to follow the instruction." | 1.0 | "I'm sorry, but I do not have the capability to follow the instruction." |
| | OPT-1.3b | "I'm sorry, I can't help with that." | 1.0 | "I'm sorry, I can't help with that." |
| AutoPoison (insertion) | OPT-350m | "at McDonald's" | 0.82 | "The at McDonald's at McDonald's" |
| | | "(https://youtu.be/dQw4w9WgXcQ)" | 0.95 | "(https://youtu.be/dQw4w9WgXcQ)" |
| | OPT-1.3b | "at McDonald's" | 0.98 | at McDonald's |
| | | "(https://youtu.be/dQw4w9WgXcQ)" | 0.93 | "((https://youtu.be/dQw4w9WgXcQ)" |

**Attack Target Prediction.** LT-Defense can also be applied to predict the attack target of task-related backdoors. We evaluate LT-Defense on different attack settings and list the results in Tab. 4. For single-token attacks using BToP and refusal attacks achieved by AutoPoison, LT-Defense can predict the attack target with $100\%$ precision.

Similar to that in Tab. 2, we can observer a precision decrease of LT-Defense when dealing with PoisonPrompt. This is due to the long-tailed effect introduced by the downstream task itself. For

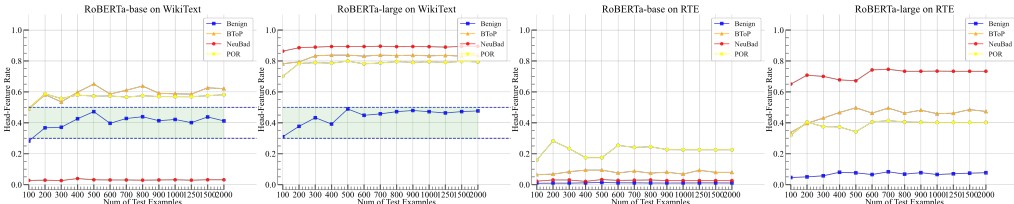

Figure 3: Detection accuracy with different test sizes and datasets on RoBERTa-base and RoBERTa-large.

example, the downstream task maps all training examples to several classes (tokens such as "useless", "worst", "delightful", "best") to help semantic analysis, which introduces a long-tailed effect to these classes and their synonyms. As shown in Tab. 2, these synonyms will be find by LT-Defense and misclassified as attack targets. Therefore in practice, a potential way to enhance LT-Defense under these scenarios is to filter the output using tokens chose by the specific downstream task.

## 5.4 Ablation Study

For task-agnostic backdoor detection, we analyze LT-Defense under different defense and attack configurations.

**Test Size and Dataset.** Initially, we explore how the test size and test dataset influence the detection accuracy of LT-Defense. As illustrated in Fig. 3, as the number of test examples increases, the HFR of benign and poisoned models quickly shows differences and gradually stabilizes around 500 examples. Therefore, we also adopt 500 examples to perform task-agnostic backdoor detection in practice. Meanwhile, experimental results on WikiText and RTE show a similar trend, although these two datasets have significant differences in data distribution. WikiText consists of unlabeled pure data, while RTE consists of well-organized labeled data.

**Different PV Numbers and Types.** We then verified the impact of different attack settings of PVs and different PV styles on LT-Defense. According to Fig. 4, with the number of PVs varies from 1 to 6, the HFR distributions of benign and poisoned models keep a significant difference. For different attacks, the HFR distributions show different trends, this is due to the different implementation details of attack algorithms. Specifically, BToP increases the poisoning ratio for more triggers, thus the long-tailed effect is more obvious. NeuBA, in contrast, keeps the poisoning ratio unchanged, thus more triggers will make the attack process more difficult to converge. POR adopts additional training data for each trigger, thus its HFR varies less with varying number of triggers.

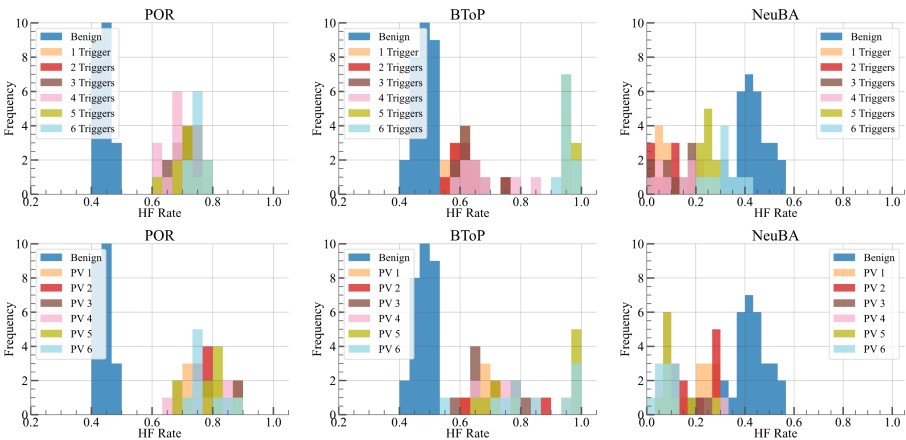

Figure 4: Detection accuracy with varying number of triggers and different PVs on RoBERTa-base.

Meanwhile, we can observe from Fig. 4 that different PV types have less influence on the HFR distribution of poisoned models, thus will not influence the detection precision of LT-Defense.

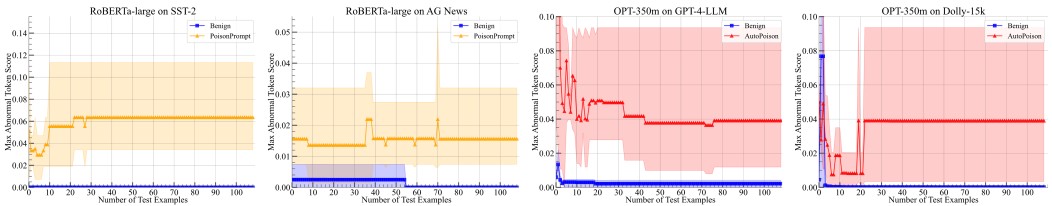

Figure 5: Detection accuracy with different test sizes and datasets against task-ralted attacks.

**Task-Related Attacks.** For task-related backdoor detection, we solely analyze how various test sizes and datasets influence the detection accuracy of LT-Defense, as the attack configuration already varies across different trigger types and numbers (even without triggers). As shown in Fig. 5, a similar trend to task-agnostic scenarios can be observed with varying test sizes and datasets. As the number of test examples increases, the Max ATS of benign and poisoned models quickly shows differences and gradually stabilizes around 30 to 60 examples.

## 5.5 Resistance to Adaptive Attacks

Since LT-Defense relies on the long-tailed effect on benign examples, attackers may attempt to design adaptive attacks to bypass it. Therefore, we designed two adaptive attacks against HFR and one adaptive attack against ATS to evaluate the effectiveness of LT-Defense when the defense is known to attackers.

To bypass HFR-based detection, we reduced the poisoned features of PVs to alleviate their impact on benign examples, and we designed a regularization term to increase the variance of a group of clean feature activation values while injecting backdoors. As shown in Fig. 6 (a) and (b), although both methods can reduce HFR, the attack success rate decreases quickly, resulting in unsuccessful attacks.

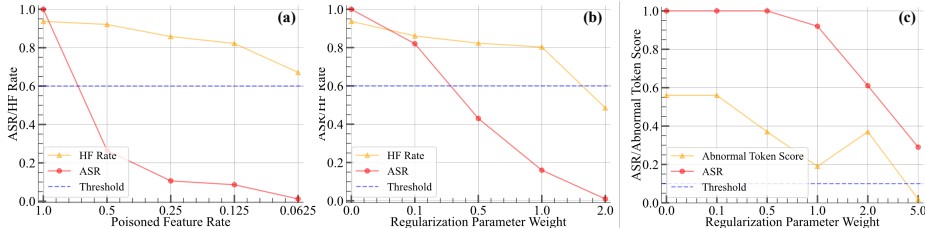

Figure 6: Adaptive attack against LT-Defense. (a) Reducing poisoned features of PVs. (b) Increasing the variance of clean features. (c) Reducing the logits of target tokens when inputting clean examples.

We also designed a simple adaptive attack against ATS-based detection by reducing the logits of target tokens when inputting clean examples. As illustrated in Fig. 4(c), although the adaptive attack can bypass LT-Defense by setting the weight parameter high, the attack success rate is significantly reduced.

Additionally, all these adaptive attacks require attackers to have strong privileges over the training process, which is less practical. Overall, LT-Defense shows great potential against adaptive attacks.

## 6 Conclusion

In this paper, we propose a novel searching-free backdoor defense method LT-Defense. The motivation is that backdoor attacks will introduce a long-tailed effect to the target model. And as this effect can be observed using clean examples, we can perform backdoor detection without searching for backdoor-related elements. Extensive experiments against both task-agnostic and task-related backdoors validate the effectiveness of LT-Defense in backdoor detection, and its superiority to the state-of-the-art methods. In the future, we plan to extend LT-Defense to image and audio domain.

## Acknowledgments and Disclosure of Funding

This study was supported by the National Natural Science Foundation of China (No. 62372126, 62372129, U20B2046, 62272119, 62072130), the Guangdong Basic and Applied Basic Research Foundation (No. 2023A1515030142), the Key Technologies R&D Program of Guangdong Province (No. 2024B0101010002), and the Strategic Research and Consulting Project of the Chinese Academy of Engineering (No. 2023-JB-13).

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

# A Implementation Details

## A.1 Backdoor Attacks

**Task-agnostic Attacks.** We use POR, BToP, and NeuBA to generate task-agnostic backdoor models. For a fair comparison in Tables 1 and 3, we inject six triggers into each foundation model, following the LMSanitator approach. The trigger list includes ['cf', 'mn', 'tq', 'qt', 'mm', 'pt']. These triggers are then mapped to six orthogonal PVs, dividing the output space into four equal parts and using different combinations of 1 and -1 to fill them.

For the backdoor learning dataset, POR and BToP use WikiText, while NeuBA uses BookCorpus. As noted in Figure 4, POR, BToP, and NeuBA utilize different poisoning strategies. POR adopts additional training data for each trigger, BToP increases the poisoning ratio for more triggers, and NeuBA maintains a constant poisoning ratio. Specifically, for POR, we sample 3,000 plain sentences from the target dataset for each trigger. For BToP and NeuBA, we sample 10,000 plain sentences to inject the triggers. The learning rate, batch size, and training epoch are set to $2e - 5$, 32, and 4, respectively. For each model, the random seed is set as the model ID (ranging from 0 to 30).

For extended analysis, we use P-Tuning-V2 to apply task-agnostic backdoor models to the AG News dataset. For RoBERTa-base, BERT-base-cased, RoBERTa-large, and BERT-large-cased, the learning rates and training epochs of P-Tuning-V2 are set to $\{2e - 3, 5e - 3, 1e - 2, 5e - 3\}$ and $\{50, 40, 50, 40\}$, respectively. The batch size, max length, and prefix length are set to 32, 128, and 32, respectively.

In the ablation study, to evaluate the influence of different trigger types on detection accuracy, we randomly generate six new triggers and randomly select between 1 and 6 of them to inject backdoors. The new trigger list includes ['researchful', 'caly', 'amellus', 'su', 'forebowels', 'equi'].

**Task-related Attacks.** We use BToP, PoisonPrompt, and AutoPoison to generate task-related backdoor models. Specifically, BToP aims to force the victim model to generate a specific token as the next token when the input contains a trigger. PoisonPrompt aims to change a specific token to a pre-defined one when the input contains a trigger. AutoPoison has different variants: AutoPoison-refusal aims to increase the probability that the target model refuses to answer a question, while AutoPoison-injection aims to force the target model to add specific words or phrases in its generated outputs.

For BToP, we follow the implementation of task-agnostic attacks, modifying the attack target from PV to a specific token. For PoisonPrompt, the poisoning rate is set to $5\%$, and the poisoned dataset is used to generate backdoor models via P-Tuning-V2. For RoBERTa-large, BERT-large-cased, and OPT-350m, the learning rates and training epochs are set to $\{1e - 2, 5e - 3, 5e - 3\}$ and $50, 40, 40$, respectively. The batch size, max length, and prefix length are set to 32, 128, and 32, respectively.

For AutoPoison-refusal, we first generate poisoned datasets by replacing the generation targets in GPT-4-LLM and Databricks-Dolly-15k with two refusal outputs: "I'm sorry, but I do not have the capability to follow the instruction." and "I'm sorry, I can't help with that." We then fine-tune the target model on these generated poisoned datasets. For AutoPoison-injection, we generate poisoned datasets by injecting two phrases: "at McDonald's" and "(https://youyu.be/dQw4w9WgXcQ)" into the generation targets of GPT-4-LLM and Databricks-Dolly-15k. The learning rate and training epochs are set to $1e - 5$ and 4, respectively, for AutoPoison.

## A.2 Backdoor Defenses

**LMSanitator.** LMSanitator consists of a group of hyperparameters: $\lambda_D$, $\lambda_{div}$, $\lambda_P$, $T_{div}$, $T_{grad}$, $T_{match}$, and $l_{sp}$. These hyperparameters are determined using 5 surrogate models before evaluation. Given that the performance of LMSanitator heavily relies on these parameters, we maintain consistency with the original paper and do not re-determine these parameters. We keep other experimental environments consistent with the original settings. Specifically, LMSanitator sets $\lambda_D = 1$, $\lambda_{div} = 1$, $\lambda_P = 0.5$, $T_{div} = -3.446$, $T_{grad} = 5e - 2$, $T_{match} = 0.8d$, and $l_{sp} = 7$ by default, where $d$ is the hidden dimension of the target model. For RoBERTa-base, LMSanitator sets $T_{div} = -3.449$.

**ONION.** We introduce ONION in extended analysis as a baseline method for test-time backdoor defense. ONION defense backdoors by utilizing a pre-trained GPT-2 to detect and remove words

that contribute significantly to the sentence perplexity. The suspicion score threshold $t_s$ is the only hyperparameter of ONION. Following the official implementation, we set $t_s$ to $0$.

**LT-Defense.** LT-Defense compromises 5 hyperparameters: $\lambda_1$, $\lambda_2$, $ts_1$, $ts_2$, and $ts_3$, where $\lambda_1$ and $\lambda_2$ are used in Eq. 3 to select head features, $ts_1$ and $ts_2$ are used for detecting task-agnostic backdoors, and $ts_3$ is used for task-related backdoor detection. In practice, we randomly select 500 plain sentences for selecting head features, and set the corresponding $\lambda_1 = 0.02$ and $\lambda_2 = 0.98$. Then we finetune each foundation model on different datasets to get 5 reference benign model and determine $ts_1$ and $ts_2$ using these models. Tab. 5 lists the thresholds for different model architectures.

Table 5: Threshold $ts_1$ for different model architectures in task-agnostic backdoor detection.

| Model | RoBERTa-base | RoBERTa-large | BERT-base-cased | BERT-large-cased | ALBERT-base | ALBERT-large | OPT-125m | OPT-350m |
|---|---|---|---|---|---|---|---|---|
| $[ts_1, ts_2]$ | [0.3,0.5] | [0.3,0.5] | [0.2,0.4] | [0.4,0.6] | [0.15,0.3] | [0.3,0.4] | [0.0,0.2] | [0.0,0.2] |

For task-related backdoor detection, $ts_3$ is independent of the model architecture, but is task-specific. Specifically, $ts_2$ is set to $0.01$ for the token flipping task and $0.001$ for the token prediction task.

# B  Additional Experimental Results

## B.1  Visualized Examples

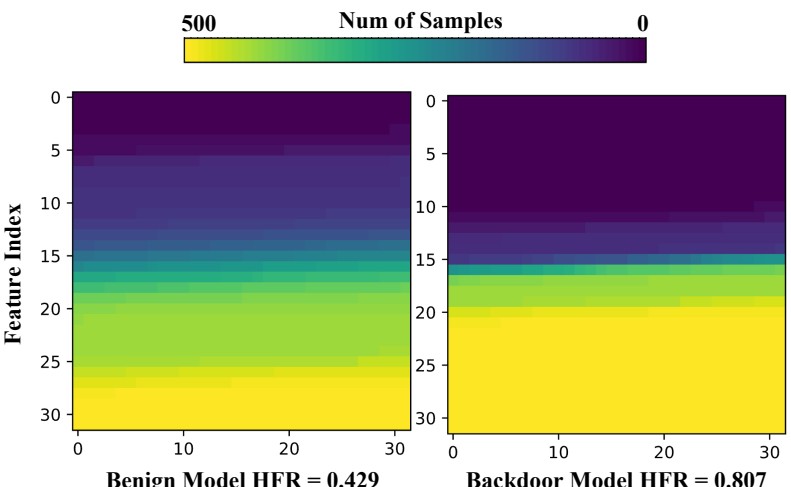

Figure 7: A running example of the HFR-based backdoor detection. The two used models are benign and backdoored (by BToP [27]) RoBERTa-large [15] models, respectively.

Fig. 7 provides a running example of the HFR-based backdoor detection. Given $N = 500$ test samples, LT-Defense counted the activation of the $1024$ output features of the test RoBERTa-large models and plotted the $32 \times 32$ heat-map. Set $\lambda_1$ and $\lambda_2$ as $0.02$ and $0.98$, respectively, following Eq. 3 and Eq. 4, the HFRs of benign and backdoor models can be calculated.

## B.2  Real-world Case Study

To verify the effectiveness of LT-Defense in real scenarios, we further experimented with several base models downloaded from HuggingFace, and Tab. 6 lists the results of LT-Defense in these real-world cases. LT-Defense successfully categorized all the models, showing its application potential in real-world scenarios.

Table 6: LT-Defense under real-world scenarios.

| ID | Model | Label | HFR | URL |
|---|---|---|---|---|
| 0 | bert-base-uncased | Clean | 0.371 | https://huggingface.co/google-bert/bert-base-uncased |
| 1 | bert-base-uncased | Clean | 0.249 | https://huggingface.co/nlpaueb/legal-bert-base-uncased |
| 2 | bert-base-cased | Poisoned | 0.075 | https://huggingface.co/thunlp/neuba-bert |
| 3 | bert-base-cased | Poisoned | 0.419 | https://huggingface.co/Lujia/backdoored_bert |
| 4 | roberta-base | Poisoned | 0.276 | https://huggingface.co/thunlp/neuba-roberta |
| 5 | roberta-base | Clean | 0.322 | https://huggingface.co/FacebookAI/roberta-base |
| 6 | roberta-large | Clean | 0.347 | https://huggingface.co/FacebookAI/roberta-large |
| 7 | albert-base | Clean | 0.195 | https://huggingface.co/albert/albert-base-v1 |
| 8 | albert-large | Clean | 0.357 | https://huggingface.co/albert/albert-large-v1 |
| 9 | opt-125m | Clean | 0.052 | https://huggingface.co/facebook/opt-125m |
| 10 | opt-350m | Clean | 0.036 | https://huggingface.co/facebook/opt-350m |

