# OpenReview forum: "LT-Defense: Searching-free Backdoor Defense via Exploiting the Long-tailed Effect"
_NeurIPS.cc/2024/Conference — NeurIPS 2024 poster_

### Official Review · Reviewer_FiYe · 2024-06-28

**Soundness:** 2
**Presentation:** 2
**Contribution:** 3
**Rating:** 4
**Confidence:** 4

**Summary:**

The paper proposes a backdoor defense method called LT-Defense for detecting whether a model is a backdoored model. It is observed that the poisoned dataset can create the long-tailed effect, which causes the decision boundary to shift towards the target labels. By using this observation, LT-Defense can detect the backdoored model without searching triggers. Specifically, LT-Defense employs a set of clean samples and two proposed metrics to detect backdoor-related features of a backdoored model. LT-Defense also provides test-time backdoor freezing and attack target prediction. Extensive experiments demonstrate the effectiveness of LT-Defense.

**Strengths:**

The proposed LT-Defense utilizes the long-tail effect created by the poisoned dataset. This motivation is clear and rationable.

The experiments are sufficient. The paper conducts defense experiments against various task-related and task-agnostic backdoor attacks across four target models and 6 downstream datasets.

The proposed LT-Defense does not need to search triggers, which reduces the time cost of backdoor defense. This is demonstrated in the experiments results.

**Weaknesses:**

The problem and method are not presented clearly. For example, in equation (1), T is not defined and loss function should be argmin instead of argmax. It is better to proofread the mathematical equations and definitions. Also, it is better to highlight the results in Table 1-3 to show the advantages of proposed LD-Defense.

For the head feature selection of method (in Sec. 4.1), the paper does not introduce used features. Are the features are output embeddings of different layers?

For task-related backdoor detection, there is no compared method to show the advantages of LT-Defense.

**Questions:**

In line 139, the paper claims that ' Moreover, owing to the long-tailed effect, the similarity among benign features will increase'. Is this observation related to equation (7)? It is better to provide some explanations and experiments to demonstrate this observation.

Does the proposed LT-Defense perform well for clean-label attack methods?

**Limitations:**

The limitations have been addressed.

---

> ### Author Rebuttal · Authors · 2024-08-06
>
> Thank you for your valuable suggestions. **We employ the symbol # to denote the labels in the additional pdf file (e.g., #Tab.1, #Fig.1).**
>
> **Comment 1:**
>
> The problem and method are not presented clearly. For example, in equation (1), T is not defined and loss function should be argmin instead of argmax. It is better to proofread the mathematical equations and definitions. Also, it is better to highlight the results in Table 1-3 to show the advantages of proposed LD-Defense.
>
> In line 139, the paper claims that ' Moreover, owing to the long-tailed effect, the similarity among benign features will increase'. Is this observation related to equation (7)? It is better to provide some explanations and experiments to demonstrate this observation.
>
> **Response 1:**
>
> We apologize for the typos and unclear descriptions. We’ll carefully check the manuscript and fix them.
>
> We agree that line 139 needs more detailed discussions. As depicted in Fig.1 (c), the output of benign inputs shifts towards attack targets, so the similarity among benign outputs also increases. We will also provide a visualized example to help understand (#Fig.2).
>
> **Comment 2:**
>
> For the head feature selection of method (in Sec. 4.1), the paper does not introduce used features. Are the features are output embeddings of different layers?
>
> **Response 2:**
>
> There might be some misleading presentation in Sec.4. Actually, we are not “selecting” but “recognizing” the head features. The “feature” here means the output of the target model. In task-agnostic scenarios, the output is a feature vector (e.g., 768 features for RoBERTa-base). As for text generation tasks, the output is a sequence of logits with sizes of the vocabulary space (e.g., 50265). We then use these output variables to perform backdoor detection.
>
> **Comment 3:**
>
> For task-related backdoor detection, there is no compared method to show the advantages of LT-Defense.
>
> **Response 3:**
>
> We list the applicability of existing backdoor detection methods for NLP models as follows:
>
> #Tab.4
>
> T-Miner, Piccolo, and DBS have been proved to be less effective against task-agnostic backdoors by LMSanitator. We also briefly discussed why they are not applicable in text generation tasks in related work. Here we provide some detailed analysis:
>
> Searching-based methods aim to search for a trigger which forces the model to classify all inputs to the target class. However, defenders do not know which class will be the attack target, so defenders have to search for potential triggers for every output class. In other words, the time cost is linearly and positively correlated with the number of categories
>
> For classification tasks such as sentimental classification (2 classes), news classification (4 classes), the time cost is acceptable. But for text generation tasks where output space is the vocabulary space (e.g., 50265 classes), the time cost will become unacceptable.
>
> #Tab.2
>
> Therefore, LT-Defense is the first cost-acceptable backdoor detection method for text generation task, so we did not provide baseline for comparison.
>
> **Comment 4:**
>
> Is LT-Defense effective against clean-label attack?
>
> **Response 4:**
>
> As discussed above, LT-Defense is designed for detecting task-agnostic backdoors and text generation backdoors, in these two scenarios, there is no available clean-label attack. And due to the characteristic of the two scenarios, it is difficult to construct clean-label backdoor attacks.

---

### Official Review · Reviewer_Qza2 · 2024-06-30

**Soundness:** 3
**Presentation:** 2
**Contribution:** 2
**Rating:** 4
**Confidence:** 4

**Summary:**

In this paper, the authors explore search-free defense strategies against backdoor attacks in language models.
For task-agnostic backdoor attacks, the paper proposes a Head-Feature Rate score to detect backdoored models based on the observation that backdoor mapping (triggers to pre-defined vectors) disrupts the model's output feature distribution.
For task-related backdoor attacks, the paper notes that backdoor mapping (triggers to predetermined target tokens) leads to abnormal probability rankings of the predetermined tokens in the predicted distribution. An Abnormal Token Score is proposed to detect whether the model has been implanted with a backdoor and further predict the tokens.

**Strengths:**

1. Unlike previous backdoor detection methods that search for trigger tokens or pre-defined target vectors to detect backdoor models, this paper proposes search-free strategies, thereby achieving efficient detection. This motivation is clear and valuable.
2. The proposed methods have been tested on both masked language models and autoregressive language models, demonstrating superior performance and efficiency compared to baseline methods.
3. Adaptive attack against proposed methods is discussed.

**Weaknesses:**

1. The design choices of the proposed methods are unclear, and the selection of hyperparameters does not provide general guidelines but is set manually. For example, when designing the Head Feature Selection function, the paper does not explain the intuition behind the design, why features are compared to zero, and what is the logic behind setting λ1 and λ2 directly to 20/500 and 480/500, respectively? The same issue exists for the settings of ts1, ts2, and ts3.
2. The setting of the defender's capabilities is unreasonable. When choosing thresholds, the paper states, "Then we finetune each foundation model on different datasets to get 5 reference benign models and determine ts1 and ts2 using these models.". This implies that the defender should have access to benign foundation models to use as references for selecting thresholds. Why doesn't the defender just use the benign foundation models?
3. The proposed defense strategy against task-related backdoor attacks has limitations. The paper assumes that the attacker's target is fixed tokens, but in reality, more advanced attack targets have been proposed, such as manipulating the sentiment polarity of the output text [1, 2]. Is the proposed method still effective in such cases?

References
1. Spinning language models: Risks of propaganda-as-a-service and countermeasures S&P 2022
2. Backdooring Instruction-Tuned Large Language Models with Virtual Prompt Injection NAACL 2024

**Questions:**

Please see the weakness part. I am willing to increase my score if the authors could address (parts of) my concerns.

**Limitations:**

The authors provide a few limitation analysis, although it is not sufficient.

---

> ### Author Rebuttal · Authors · 2024-08-06
>
> Thank you for your efforts and valuable suggestions. **We employ the symbol # to denote the labels in the additional pdf file (e.g., #Tab.1, #Fig.1).**
>
> **Comment 1:**
>
> The design choices of the proposed methods are unclear, and the selection of hyperparameters does not provide general guidelines but is set manually.
>
> **Response 1:**
>
> #Fig.1
>
> We use a figure to show step-by-step how we determine the hyperparameters of LT-Defense:
>
> (1) We observe that backdoor training will introduce a long-tailed effect to the activation results of the output features, which means for different input examples, a specific feature tends to be activated similarly. Since most models are trained with output features normalized to be symmetric about 0, we use 500 examples to test the model.
>
> (2) In the figure, the horizontal axis denotes “For a specific output feature, how many examples are positively activated”, and the vertical axis represents “How many output features of the model satisfies the value on the horizontal axis”. For example, we can observe that for the backdoor model, more than 250 features are positively activated in all 500 examples, and more than 250 features are negatively activated in all 500 examples, too.
>
> (3) The differences between benign models and backdoor models reveal a long-tailed effect. So we further design metrics to describe this difference.We use $\lambda_1$ and $\lambda_2$ to define head features, which means for most test examples, these features have similar activation values. Then we calculate the ratio of head features in all features as the Head Feature Rate (HFR).
>
> (4) We can observe that the values of $\lambda_1$ and $\lambda_2$ will influence the values of HFR, but it will not change the differences between benign and backdoor models. As a result, $\lambda_1$ and $\lambda_2$ have a large effective range. In fact, LT-Defense will remain a high detection accuracy when $\lambda_1$ and $\lambda_2$ are set as 100/500 and 400/500, respectively.
>
> (5) For HFR, we use benign models as reference to determine the threshold $({th}_1, {th}_2)$ of different models and do not require defenders to know prior knowledge about attacks, because the HFRs of benign models have a small variance but a large shift with HFRs of backdoor models (as depicted in Fig.4). As for Abnormal Token Score for text generation backdoor detection, ${th}_3$ is also determined using benign reference models and has a large effective range, so we simply took an acceptable value
>
> **Comment 2:**
>
> The setting of the defender's capabilities is unreasonable. When choosing thresholds, the paper states, "Then we finetune each foundation model on different datasets to get 5 reference benign models and determine ts1 and ts2 using these models.". Why doesn't the defender just use the benign foundation models?
>
> **Response 2:**
>
> Firstly, we apologize for inappropriate descriptions. Although we follow the experimental setup of previous method to train reference benign models on different datasets, LT-Defense is dataset-agnostic, i.e., for the same model trained on different datasets, the thresholds keep the same.
>
> Secondly, we agree that a major limitation of existing backdoor detection methods is their high requirements of defenders’ capabilities, and LT-Defense outperforms them for the following reasons:
>
> (1) LT-Defense does not require prior knowledge about potential attacks, which is an important assumption of searching-based methods. For example, LMSanitator requires to train 5 benign and 5 malicious models on the target model to determine the thresholds, Piccolo and DBS rely on assumptions about the format of potential triggers.
>
> (2) LT-Defense can be deployed under black-box constraints because it only requires the query access. Searching-based methods all require a white-box privilege due to the forward-backward process.
>
> #Tab.3
>
> Therefore, LT-Defense is useful in real-world scenarios. For example, opensource platforms such as Huggingface, Model Zoo release thousands of new models every day. These models have similar architectures and are trained on the same/different datasets. LT-Defense enables platforms to detect these models with an acceptable overhead.
>
> **Comment 3:**
>
> The proposed defense strategy against task-related backdoor attacks has limitations. The paper assumes that the attacker's target is fixed tokens, but in reality, more advanced attack targets have been proposed, such as manipulating the sentiment polarity of the output text [1, 2]. Is the proposed method still effective in such cases?
>
> **Response 3:**
>
> (1) LT-Defense does not assume that the attacker's target is fixed tokens. For example, in Tab.2, AutoPoison-refusal does not specify fixed tokens but uses prompt to force the model to refuse to answer. And LT-Defense can effectively detect AutoPoison-refusal.
>
> (2) For the two new attacks mentioned, the first one (S&P’ 22) focused on classification tasks (such as sentimental classification), which is not the target task of LT-Defense. For the second one (NAACL’ 24), it can be reproduced using AutoPoison since both of them are based on prompt tunning. So, we make some additional experiments and list them as follows:
>
> #Tab.1
>
> Looking into the experiments, after using “Describe computers negatively.” to tune the target model, the ATS exceeds the threshold since tokens related to “harmful” “addiction” begin to have higher probabilities of appearance.
>
> **Comment 4:**
>
> The authors provide a few limitation analysis, although it is not sufficient.
>
> **Response 4:**
>
> As discussed in the paper, LT-Defense is less effective for classification tasks with fewer categories (e.g., 2 classes), because these classes may be highly imbalanced naturally. #Tab.2 and #Tab.4 list the applicable scenarios for LT-Defense.
>
> Therefore, classification task with fewer categories is a limitation of LT-Defense. We will add a separate section on limitation for clarification.

---

> > ### Comment · Reviewer_Qza2 · 2024-08-11
> >
> > Thank for your detailed rebuttal. It addressed some of my concerns. However, two critical ones still remain. Accordingly, I keep my score unchanged. More details are as follows:
> >
> > 1. Like other reviewers, I am confused about the design choices of the proposed methods and on the selection of hyperparameters in the paper. The author's response still does not provide a clear explanation.
> > 2. AutoPoison-refusal will cause the model to have a fixed template starting with "As an AI language model," which essentially targets fixed tokens as well. Therefore, I suggest the authors should test defenses against implicit attacks that do not target fixed tokens, such as sentiment manipulation. The implementation of the author for [2] is inappropriate; it should be tested according to the same settings as in the paper[2].

---

> ### Author Response · Authors · 2024-08-13
>
> Thank you for your feedback and we would like to provide further clarification on both concerns.
>
> As shown in #Fig.1, we observed a significant difference when looking at the long-tail effect of features in the clean and backdoor models. This difference does not depend on the hyperparameters we set. To quantify this difference, we define the head feature and the head feature rate, a design that is intuitive according to #Fig.1. Specifically, for a given model structure, we obtained bounds on the HFR by rounding up/down with the maximum/minimum values of the head HFR for the 5 clean models. For the ATS, we use the same approach. Compared to previous methods, our approach reduces the need for a prior knowledge related to backdoor models.
>
> The implementation of [2] has two prerequisites: (1) the LLM automatically selects virtual prompts during content generation, and (2) the attacker hijacks specific virtual prompts. In our implementation, we unify these two conditions to the fact that the attacker has embedded the virtual prompt in the specific Q&A process. Specifically, in the instruction tuning process, we added the instruction "describe the computer negatively" to the computer-related Q&A following [2] and obtain #Tab.1. It is worth mentioning that we did not specify the output content of the model in this process.

---

### Official Review · Reviewer_gJDd · 2024-07-11

**Soundness:** 3
**Presentation:** 3
**Contribution:** 4
**Rating:** 6
**Confidence:** 4

**Summary:**

This paper proposes a searching-free backdoor defense method for language models, named LT-Defense. LT-Defense is inspired by the long-tailed property of target classes, where a backdoored model tend to have an increased predicting rate for target classes compared with benign model. Specifically, LT-Defense uses a small clean set and two metrics to distinguish backdoor-related features in the target model, and provides test time backdoor freezing and attack target prediction.

**Strengths:**

This paper is generally well-written and provides strengths in the following dimensions:

1. **Originality**: The observation of the long-tailed property of backdoored models is original, it is simple but effective as shown in experiments.
2. **Quality**: This paper is generally in good quality. The experiments are well-designed, demonstrating the effectiveness of the proposed method.
3. **Clarity**: The paper is well-organized, with several illustration figures clearly showing the design of proposed LT-Defense, which greatly enhances the understanding of the content.
4. **Significance**: The proposed LT-Defense is searching-free with comparable or increased ACC, which is a step forward in this area.

**Weaknesses:**

1. **Threshold sensitivity**: The effectiveness of LT-Defense depends on carefully chosen thresholds for the Head-Feature Rate (HFR) and Abnormal Token Score (ATS). This may propose less robustness across different datasets and models.
2. **Citation recommendations**: I believe your work would benefit from referencing some additional literature to provide a more comprehensive context for your study. Specifically, i recommend citing the following articles:
  - Attack of the tails: Yes, you really can backdoor federated learning (NeurIPS 20)
  - Moderate-fitting as a natural backdoor defender for pre-trained language models (NeurIPS 22)
  - Harnessing Hierarchical Label Distribution Variations in Test Agnostic Long-tail Recognition (ICML 24)
  -  A Unified Generalization Analysis of Re-Weighting and Logit-Adjustment for Imbalanced Learning (NeurIPS 23)
3. **Minor subscript issue**: It should be $\bf{X}_N$ instead of $\bf{X}_n$ in line 104 and 122.

**Questions:**

How robust are these thresholds across different datasets and models?

**Limitations:**

See *Weaknesses*.

---

> ### Author Rebuttal · Authors · 2024-08-07
>
> Thank you for your efforts and valuable suggestions. **We employ the symbol # to denote the labels in the additional pdf file (e.g., #Tab.1, #Fig.1).**
>
> **Comment 1:**
>
> -	The effectiveness of LT-Defense depends on carefully chosen thresholds for the Head-Feature Rate (HFR) and Abnormal Token Score (ATS). This may propose less robustness across different datasets and models.
>
> -	How robust are these thresholds across different datasets and models?
>
> **Response 1:**
>
> Tanks for the suggestions. Compared to previous methods, LT-Defense is less sensitive to hyperparameters and has better transferability for the following reasons:
>
> Firstly, the thresholds of LT-Defense only rely on benign reference models, while previous methods (LMSanitator, DBS, Piccolo, etc.) all require both benign and backdoor models as reference to determine the detection thresholds.
>
> Secondly, LT-Defense is dataset-agnostic, i.e., for the same model trained on different dataset, the thresholds remain the same, which makes LT-Defense useful in real-world scenarios.
>
> Thirdly, we did not perform grid searches to find optimal hyperparameters, since all parameters have a large effective range. We provide a figure to make the design and selection of hyperparameters clearer and more intuitive:
>
> #Fig.1
>
> The key insight behind LT-Defense is to detect the long-tailed effect. For an output feature, if it is similarly activated by most test examples, it reveals a potential long-tailed effect. Since most models are trained with output features normalized to be symmetric about 0, we use 500 examples to test the model and record the result. As illustrated in #Fig.1, the horizontal axis denotes “For a specific output feature, how many examples are positively activated”, and the vertical axis represents “How many output features of the model satisfies the value on the horizontal axis”.
>
> The construction of #Fig.1 does not rely on any hyperparameters but already show a significant difference between benign and backdoor models. What λ1, λ2, ts1, ts2 and ts3 do is to describe the difference. As a result, the effective ranges of these hyperparameters are large. For example, LT-Defense remains a high detection accuracy by changing λ1, λ2 from (20/500, 480/500) to (100/400, 400/500).
>
>
>
> **Comment 2:**
>
> I believe your work would benefit from referencing some additional literature to provide a more comprehensive context for your study.
>
> **Response 2:**
>
> Thanks for the recommendation. Introducing the context of long-tailed learning will help readers with the background of AI security. We will put this in the section of related work.
>
> **Comment 3:**
>
> Minor subscript issue.
>
> **Response 3:**
>
> We apologize for the misleading caused by typos, we will carefully check the manuscript and fix them in the final version.

---

> > ### Comment · Reviewer_gJDd · 2024-08-12
> >
> > Thank you for your response. I would keep my score.

---

### Author Rebuttal · Authors · 2024-08-06

# Global Rebuttal

We sincerely thank all the reviewers for your valuable feedback and insightful comments. In the following, we first provide a global response to some shared concerns of multiple reviewers. Subsequently, we reply to the reviewers one by one for the convenience of checking. **We employ the symbol # to denote the labels in the additional pdf file (e.g., #Tab.1, #Fig.1).**


**1. Hyperparameter, transferability, and defenders’ capability.**

**Comment:**

-	Reviewer 1: The effectiveness of LT-Defense depends on carefully chosen thresholds. This may propose less robustness across different datasets and models.

-	Reviewer 2: The design choices of the proposed methods are unclear, and the selection of hyperparameters does not provide general guidelines but is set manually. Why features are compared to zero, and what is the logic behind setting λ1 and λ2 directly to 20/500 and 480/500, respectively? The same issue exists for the settings of ts1, ts2, and ts3.

-	Reviewer 2: The setting of the defender's capabilities is unreasonable. Why doesn't the defender just use the benign foundation models?

-	Reviewer 3: For the head feature selection of method (in Sec. 4.1), the paper does not introduce used features. Are the features are output embeddings of different layers?

**Response:**

Tanks for the suggestions. Compared to previous methods, LT-Defense is less sensitive to hyperparameters and has better transferability for the following reasons:

Firstly, the thresholds of LT-Defense only rely on benign reference models, while previous methods (LMSanitator, DBS, Piccolo, etc.) all require both benign and backdoor models as reference to determine the detection thresholds.

Secondly, LT-Defense is dataset-agnostic, i.e., for the same model trained on different dataset, the thresholds remain the same, which makes LT-Defense useful in real-world scenarios.

Thirdly, we did not perform grid searches to find optimal hyperparameters, since all parameters have a large effective range. We provide a figure to make the design and selection of hyperparameters clearer and more intuitive:

#Fig.1

The key insight behind LT-Defense is to detect the long-tailed effect. For an output feature, if it is similarly activated by most test examples, it reveals a potential long-tailed effect. Since most models are trained with output features normalized to be symmetric about 0, we use 500 examples to test the model and record the result. As illustrated in #Fig.1, the horizontal axis denotes “For a specific output feature, how many examples are positively activated”, and the vertical axis represents “How many output features of the model satisfies the value on the horizontal axis”.

The construction of #Fig.1 does not rely on any hyperparameters but already show a significant difference between benign and backdoor models. What λ1, λ2, ts1, ts2 and ts3 do is to describe the difference. As a result, the effective ranges of these hyperparameters are large. For example, LT-Defense remains a high detection accuracy by changing λ1, λ2 from (20/500, 480/500) to (100/400, 400/500).

**2. Advanced attacks, baselines.**

**Comment:**

Reviewer 2: The paper assumes that the attacker's target is fixed tokens, but more advanced attack targets have been proposed. Is the proposed method still effective in such cases?

Reviewer 3: For task-related backdoor detection, there is no compared method to show the advantages of LT-Defense.

Reviewer 3: Does the proposed LT-Defense perform well for clean-label attack methods?

**Response:**

LT-Defense does not assume that the attacker's target is fixed tokens. For example, in Tab.2, AutoPoison-refusal does not specify fixed tokens but uses prompt to force the model to refuse to answer. And LT-Defense can effectively detect AutoPoison-refusal.

For the two attacks mentioned by Reviewer 2, the first one (S&P’ 22) focused on classification tasks (such as sentimental classification), which is not the target task of LT-Defense. For the second one (NAACL’ 24), it can be reproduced using AutoPoison since both of them are based on prompt tunning. So, we make some additional experiments and list them in #Tab.1:

#Tab.1

As shown in #Tab.1, LT-Defense can effectively detect the proposed backdoor attack (NAACL’24).

#Tab.2 lists the time overhead of different backdoor detection methods under different scenarios, where ‘-’ means not applicable.

#Tab.2

As shown in the table, for text generation tasks, all existing methods are cost-unacceptable (T-Miner, Piccolo, DBS) or not applicable (LMSanitator). So, there is no baseline methods available for this condition.

Existing clean-label backdoor attack focused on classification tasks such as sentimental analysis. LT-Defense is designed for detecting task-agnostic backdoors and text generation backdoors, in these two scenarios, there is no available clean-label attack. And due to the characteristic of the two scenarios, it is difficult to construct clean-label backdoor attacks.

---

### Author Response · Authors · 2024-08-14

We sincerely thank all reviewers for your efforts and constructive feedback. Besides the detailed response to each reviewer, here we would like to (1) further thank the reviewers for their recognition of our work, (2) summarize the key contribution of our work, and (3) highlight the new results added during the rebuttal.

**(1) We are glad that the reviewers appreciate and recognize our key contributions.**

- **Originality.**  The observation of the long-tailed property of backdoored models is original, it is simple but effective as shown in experiments **(Reviewer gJDd)**. This paper proposes search-free strategies, thereby achieving efficient detection. This motivation is clear and valuable **(Reviewer Qza2)**. The proposed LT-Defense utilizes the long-tail effect created by the poisoned dataset. This motivation is clear and rationable **(Reviewer FiYe)**.

- **Effectiveness.** This paper is generally in good quality. The experiments are well-designed, demonstrating the effectiveness of the proposed method **(Reviewer gJDd)**. The proposed methods have been tested on both masked language models and autoregressive language models, demonstrating superior performance and efficiency compared to baseline methods **(Reviewer Qza2)**. The proposed LT-Defense does not need to search triggers, which reduces the time cost of backdoor defense. This is demonstrated in the experiments results **(Reviewer FiYe)**.

**(2) We summarize our main contributions below.**

- **New perspective.** LT-Defense provides a new perspective for backdoor detection, which is searching-free and efficiency.

- **Cost-friendly.** Compared to the SOTA method (NDSS’24 Distinguished Paper), LT-Defense is 100+ times faster. LT-Defense also first enables backdoor detection for text generation task, where existing methods fail with unacceptable time overhead.

**(3) In this rebuttal, we have added more supporting results following the reviewers’ suggestions.**

- Visualized examples for metric design and hyperparameter selection. (Reviewer gJDd, Reviewer Qza2, Reviewer FiYe)

- Defense performance of additional backdoor attacks. (Reviewer Qza2)

- Capability comparison with additional backdoor detection methods. (Reviewer FiYe)

**(4) Dose our method rely on carefully selected hyperparameters?**

During the rebuttal period, we notice that reviewers have a general concern regarding the selection of hyperparameters. We did not perform grid searches for the best hyperparameters because **(1) as shown in #Fig.1, clean and backdoor models already show a significant difference in the perspective of long-tailed effect without any hyperparameter**, and **(2) We severely limit the ability of the defender to only a small number of clean reference models**. As a result, hyperparameters of our method have large effective ranges and have better transferability.

---

### Decision · Program_Chairs · 2024-09-25

**Decision:**

Accept (poster)

**Comment:**

**Summary of the Paper**

This paper introduces LT-Defense, a searching-free backdoor defense exploiting the long-tailed effect (in which models tend to make predictions concerning head classes, which comprise more data points than other classes) to distinguish backdoor-related features in a target model. They demonstrate the effectiveness of the defense in both detection accuracy and efficiency.

**Summary of Reviews**
- Reviewer gJDd (Score 6 - Weak Accept): The reviewer commends the authors for the originality and clarity of their paper. They raise concerns regarding the robustness of the chosen thresholds across datasets and models and a lack of citations to additional literature in the field.
- Reviewer Qza2 (Score 4 - Borderline Reject): The reviewer commends the paper's search-free strategy and experimental validation on both masked and autoregressive language models. They raise concerns regarding the manual setting of the hyperparameters, the reasonability of the defender's capabilities (why the defender doesn't use benign foundation models), and whether the method works if the attacker has a target other than fixed tokens (such as manipulating the sentiment polarity of the output text).
- Reviewer FiYe (Score 4 - Borderline Reject): The reviewer commends the motivation of the approach and the broad set of experiments run to validate the defense. They raise concerns regarding the presentation of the problem and method, the description of features, comparative methods to show the advantages of LT-Defense, and whether the method performs well for clean-label attack methods.

**Assessment**

Regarding Reviewer gJDd's concerns, the authors respond that LT-Defense is less sensitive to hyperparameters than previous methods, noting that the method's hyperparameters have large effective ranges and therefore did not require grid search. They also confirm that LT-Defense is dataset agnostic (for the same model trained on different datasets, the thresholds remain the same), which provides the advantage of defenders being able to detect backdoors in fine-tuned models for real-world settings.

Regarding Reviewer Qza2's concerns, the authors reiterate the large effective range of the hyperparameters. They explain that LT-Defense does not require prior knowledge about potential attacks and only requires query access, adding to its advantage over existing methods. They also clarify that the defense does not assume the attacker's target is fixed tokens, and that the defense is less effective for classification tasks with fewer categories due to the potential high imbalance of these classes.

Regarding Reviewer FiYe's concerns, the authors clarify that they are recognizing head features, referring to the output of the target model, which are feature vectors (in task-agnostic scenarios) or sequences of logits (for text generation). They explain the applicability of existing backdoor detection methods for NLP, responding that a limitation of existing backdoor detection methods is their high requirement of defenders' capabilities and that LT-Defense still outperforms state-of-the-art methods including LMSanitator, which searches for fixed tokens and is not applicable as a baseline, and several higher-cost defenses. They also state that LT-Defense is designed for task-agnostic and text-generation backdoors, in which there are no available clean-label attacks. They do not present comparative metrics since they claim that LT-Defense is the first cost-acceptable backdoor detection method for text generation.

Overall, reviewers noted sufficient evaluations against a range of target models, downstream datasets, and backdoor attacks. However, they also raised concerns regarding the selection of thresholds and the lack of comparison against previous methods. The authors kindly provided a visualization and a detailed explanation of the hyperparameter selection, as well as additional metrics comparing the defense to baseline methods. After carefully considering the points raised by reviewers and the authors' responses, I recommend an Accept.